# Learning Equivalence Classes of Bayesian Network Structures with GFlowNet

## Abstract

Understanding the causal graph underlying a system is essential for enabling causal inference, particularly in fields such as medicine and genetics. Identifying a causal Directed Acyclic Graph (DAG) from observational data alone is challenging because multiple DAGs can encode the same set of conditional independencies, collectively represented by a Completed Partially Directed Acyclic Graph (CPDAG). Effectively approximating the CPDAG is crucial because it facilitates narrowing down the set of possible causal graphs underlying the data. We introduce CPDAG-GFN, a novel approach that uses a Generative Flow Network (GFlowNet) to learn a posterior distribution over CPDAGs. From this distribution, we can sample to create a set of plausible candidates that approximate the ground truth. This method focuses on sampling high-reward CPDAGs, with rewards determined by a score function that quantifies how well each graph fits the data. Additionally, it incorporates a sparsity-preferring filtering mechanism to enhance the produced set of CPDAGs. Experimental results on both simulated and real-world datasets demonstrate that CPDAG-GFN performs competitively with state-of-the-art methods for learning CPDAG candidates from observational data.

## 1 Introduction

Causal graphs produced from observational data are highly sought after, because knowing the causal dag underlying a system enables counterfactual reasoning, allows predictions about the system, and may enhance the generalizability of machine learning models (Schölkopf et al., 2021).

A causal structure is typically represented by a Directed acyclic graph (DAG). However, a significant challenge arises. In real-world scenarios, causal graphs often cannot be identified from observational data alone. However, a significant challenge arises because multiple DAGs can encode the same set of conditional independencies, often making it impossible to distinguish from observational data alone which DAG represents the true causal structure (Koller & Friedman, 2009)[1]. Consequently, DAGs that encode the same set of conditional independencies can be grouped into a single class known as a Markov Equivalence Class (MEC) (Castelletti et al., 2018). When working with observational data, all DAGs in the same MEC are equally plausible representations of the causal structure. Thus, we generally can at most learn the causal graph up to its MEC from observational data alone (Chickering, 2002a; Koller & Friedman, 2009).

A MEC is represented by a Completed Partially Directed Acyclic Graph (CPDAG, Castelletti et al., 2018, see Figure 1). Unlike DAGs, CPDAGs can explicitly capture the uncertainty inherent in causal discovery by using undirected edges to represent ambiguous causal directions. This representation not only reflects the limits of observational data but also helps identify which edges to target for interventions to resolve causal ambiguities (Pearl, 2009; Brouillard et al., 2020).

---

[1]While it is commonly the case that DAGs can be learned only up to their Markov Equivalence Class (MEC) from observational data alone, there are special cases under certain conditions where exact identification is possible. For detailed examples, see Shimizu et al. (2006); Hoyer et al. (2008); Peters & Bühlmann (2014).

Popular methods, such as the PC algorithm (Spirtes et al., 2001), typically identify only a single CPDAG from observational data, potentially overlooking other promising candidates. Like all algorithms, these methods rely on certain assumptions for optimal performance, some of which are challenging to meet in practice. For instance, the PC algorithm often requires unrealistic conditions, such as infinite data or perfect oracles for independence tests - conditions that are difficult to meet in practice. Consequently, these methods may fail to accurately identify the true CPDAG. If the true causal structure falls into a different equivalence class than the one predicted by the model, sticking to one class may overlook potential better-fitting models. Given these limitations, it is preferable to employ an algorithm that can return multiple CPDAGs from observational data.

One approach is to adopt a Bayesian method to obtain a posterior distribution over all possible CPDAGs, allowing for the sampling of multiple CPDAGs that could explain the data. Our aim, on the other hand, is to produce a set of plausible CPDAG candidates by generating a distribution directly over CPDAGs. In this spirit, we propose to combine a Bayesian approach and a filtering mechanism. This approach differs fundamentally from methods like DAG-GFN, introduced by Deleu et al. (2022).

Deleu et al. (2022) introduce a novel approach called DAG-GFN, which uses GFlowNet with a uniform prior to approximate the posterior distribution over DAGs from observational data, where probabilities are approximately proportional to the reward. The reward is determined by a score function that measures how well a DAG fits the observations. The better the fit, the higher the probability of sampling that DAG. CPDAGs can be obtained by converting the sampled DAGs to CPDAGs.

However, DAG-GFN's reliance on a score function and a uniform prior to approximate a posterior distribution over DAGs can present significant drawbacks. More concretely, the posterior distribution trained by DAG-GFN is approximately proportional to the reward induced by the score function, meaning that the learned distribution is heavily influenced by this function. If the score function assigns high scores to many DAGs, including those not representing the ground truth and potentially even those with higher scores than the ground truth (a scenario likely to occur in settings with insufficient observational data (Friedman & Koller, 2013)), the posterior distribution may become skewed, diverging significantly from the true distribution underlying the data. In other words, it is likely that our prior over graphs matters. In DAG-GFN, the choice of a uniform prior treats all DAGs as equally likely, not incorporating any knowledge that could guide the model toward a more accurate distribution. This can lead to poor approximations of the dataset's underlying distribution, resulting in samples that may not adequately reflect the true graph.

Taking the above limitation into account, we introduce a new algorithm called CPDAG-GFN, which uses GFlowNet to produce sets of CPDAG candidates from observational data. Unlike DAG-GFN, which searches within the DAG space, our approach operates directly in the CPDAG space, enabling direct CPDAG sampling. Moreover, instead of relying on the posterior to produce a final set of candidates, as in DAG-GFN, we rely on the posterior as an amortized sampler from which we can select an ideal set of candidates. In particular, our approach yields relatively high top-K scoring CPDAGs from this amortized sampling. This is possible because GFlowNets can be seen as amortized samplers capable of exploring multiple high-reward states (e.g., scores) during training. However, as mentioned above, relying solely on scores to prioritize graphs may lead to discrepancies with the ground truth, as high-scoring graphs might significantly deviate from it. To address this, we refine the sampled candidate graphs by incorporating additional knowledge into our CPDAG-GFN algorithm. One can think of this as imposing a prior belief into our algorithm, though not in the Bayesian sense. Specifically, we enhance our algorithm by applying a heuristic filter, removing the least common edges among the sampled graphs. This additional step is based on the conjecture that the top-K graphs sampled from GFlowNet often share common edge features likely present in the true graph, helping to align the top-K scoring graphs more closely with the actual CPDAG underlying the data.

The contributions of this paper are as follows: We introduce a novel algorithm named CPDAG-GFN [2], designed to learn multiple CPDAG candidates from observational data. We evaluate our method using both synthetic and real-world datasets and demonstrate that it performs competitively with state-of-the-art methods.

---

[2]Code can be found at `https://anonymous.4open.science/r/for_TMLR2-ED38`

## 2 Preliminaries

In this section, we review the concepts relevant to our proposed method, CPDAG-GFN.

### 2.1 Bayesian networks, Markov Equivalence Class, CPDAGs

A **Bayesian network** (Pearl, 2009; Koller & Friedman, 2009) is a probabilistic graphical model represented by a DAG over a set of random variables $X_1, \ldots, X_n$. Each variable $X_i$ is associated with a collection of conditional distributions given its parent nodes, denoted as $\text{Pa}(X_i)$ [3]. The dependency structure of a Bayesian network leverages the *Markov Property*, which asserts that each variable $X_i$ is conditionally independent of its non-descendants in the graph, given its parents $\text{Pa}(X_i)$. This property allows us to factorize the joint distribution of the network into the product of conditional probabilities for each node given its parents (Jin et al., 2023):

$$P(X_1, X_2, \ldots, X_n) = \prod_{i=1}^{n} P(X_i \mid \text{Pa}(X_i)) \tag{1}$$

**Markov Equivalence Classes (MECs):** DAGs that encode the same conditional independencies are Markov equivalent and are said to belong to the same MEC. These DAGs induce the same joint distribution (Jin et al., 2023). DAGs that are Markov equivalent share the same skeleton [4] and v-structures (see below for definition).

**CPDAGs:** A MEC is represented by a CPDAG, also known as an *essential graph* (Castelletti et al., 2018). A CPDAG is a type of partially directed graph that may consist entirely of directed edges, entirely of undirected edges, or a combination of both. It primarily consists of the following three types of edges:

- *Directed edge:* If an edge $x \to y$ appears in every DAG in the MEC, then the CPDAG contains a directed edge $x \to y$.
- *Undirected edge:* If the edges $x \to y$ and $y \to x$ each appear in at least one DAG in the MEC, then the CPDAG contains an undirected edge between $x$ and $y$.
- *V-structure:* If an ordered triple of nodes $(x, y, z)$ forms a configuration where $x \to y \leftarrow z$ and $x$ and $z$ are not connected by any edge, then this configuration is classified as a v-structure (Pearl, 2009).

The following theorem provides necessary and sufficient conditions for a graph to be the CPDAG of some MEC, which is essential for defining the search space for our CPDAG-GFN algorithm in Section 3.2.1.

**Theorem 1** (Andersson et al. (1997))**.** *A graph $G$ is a CPDAG if and only if it satisfies all of the following four conditions:*

*(a) $G$ contains no directed cycle.*
*(b) Every chain component of $G$ is chordal.*
*(c) The graph $a \to b - c$ does not occur as an induced subgraph of $G$.*
*(d) Every directed edge $a \to b$ in $G$ is strongly protected in $G$.[5]*

Note that not every partially directed graph qualifies as a CPDAG of some MEC, as indicated in the theorem above.

### 2.2 GFlowNet

Generative flow networks (Bengio et al., 2021; 2023), or GFlowNets, were introduced as a framework to learn to sample from an unnormalized density function, typically referred to as the reward $R(s)$ in GFlowNet literature, by decomposing the generative process in a trajectory of constructive steps. GFlowNets work by modelling the *flow* that goes through the network representing the space of possible constructions, accounting

---

[3]$\text{Pa}(X_i)$ represents the collection of values of all parent nodes; for a node without parents, this set is empty.
[4]The skeleton of a DAG refers to the undirected graph obtained by ignoring the direction of all edges in the DAG.
[5]Refer to Andersson et al. (1997) for the definition of 'strongly protected' and a review of the terms in (b) above.

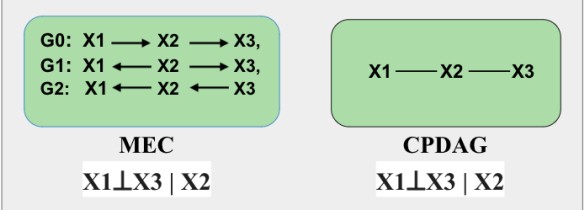

| # variables | # MECs | MECs/DAGs |
|---|---|---|
| 1 | 1 | 1.00000 |
| 2 | 2 | 0.66667 |
| 3 | 11 | 0.44000 |
| 4 | 185 | 0.34070 |
| 5 | 8782 | 0.29922 |
| 6 | 1067825 | 0.28238 |
| 7 | 312510571 | 0.27443 |
| 8 | 212133405200 | 0.27068 |
| 9 | 32626056291213 | 0.26888 |
| 10 | 111890205449597514 | 0.26799 |

Figure 1: Illustration of a MEC and its corresponding CPDAG for 3 variables. The MEC consists of multiple DAGs (i.e. $G_0$, $G_1$, $G_2$) that share identical conditional independencies, represented by $X_1 \perp X_3 \mid X_2$. The CPDAG is a single graph that compactly represents these DAGs, encapsulating the same conditional independencies.

Figure 2: The number of CPDAGs as a function of the number of variables. The column labeled 'CPDAG/DAG' represents the ratio between the number of CPDAGs and the number of DAGs for each variable count (Gillispie & Perlman, 2002).

for all possible construction orders of an object. We introduce the framework for a discrete setting, but continuous settings are also possible (Lahlou et al., 2023).

In a GFlowNet the state space is defined by a pointed DAG, which we denote $\mathcal{H} = (\mathcal{S}, \mathcal{A})$, with a unique initial state $s_0 \in \mathcal{S}$ and some terminal states $\mathcal{X} \subseteq \mathcal{S}$ on which $R : \mathcal{X} \to \mathbb{R}_{\geq 0}$ the reward function is defined. We define the exact state space we use in Section 3.2.1, but note that the GFlowNet DAG is distinct from the (CP)DAGs presented above; in fact, states within $\mathcal{H}$ are themselves (CP)DAGs. Elements of the action space $(s \to s') \in \mathcal{A}$ denote valid constructive steps, such as adding a directed edge in a CPDAG (see Figure 3), or may represent the action of ending generation. GFlowNets define a forward policy used for sampling objects, $P_F(s'|s)$, a backward policy $P_B(s|s')$ (a policy on the reverse Markov decision process, i.e., a model giving a distribution over the parents of any state), and an estimate of the partition function $Z$ representing the sum of all rewards–and the total flow when interpreting $\mathcal{H}$ as a network. The policy $P_F$ determines a terminating distribution $P_F^\top$ over $\mathcal{X}$, which is the marginal distribution over the final states of trajectories sampled following $P_F$ (i.e., those at which the termination action is taken).

GFlowNets are trained by driving a model to respect constraints which preserve the flow within $\mathcal{H}$. One such set of constraints are the *trajectory balance* constraints (Malkin et al., 2022), whereby for any trajectory $\tau = (s_0 \to s_1 \to ... \to s_n = x)$:

$$Z \prod_{i=1}^{n} P_F(s_{i+1}|s_i) = F(s_n) \prod_{i=1}^{n-1} P_B(s_i|s_{i+1}) \tag{2}$$

where $F(s_n) = R(x)$ is the flow of the terminal (sink) state $x$. With the above constraints satisfied, sampling transitions starting from $s_0$ and using $P_F$ guarantees that the marginal terminating distribution $P_F^\top(x) \propto R(x)$.

In the present work we parameterize $\log Z_\theta$ and $P_F(\cdot|\cdot; \theta)$, use a uniform $P_B$, and apply the trajectory balance objective (Malkin et al., 2022), which for a trajectory $\tau = (s_0 \to s_1 \to ... \to s_n = x)$ is:

$$\mathcal{L}_{\text{TB}}(\tau) = \left( \log \frac{Z_\theta \prod_{i=0}^{n-1} P_F(s_{i+1}|s_i; \theta)}{R(x) \prod_{i=0}^{n-1} P_B(s_i|s_{i+1})} \right)^2. \tag{3}$$

When training with GFlowNet objectives, one samples trajectories from some behaviour policy (which may either coincide with $P_F$ – *on-policy training*, as done in this paper – or use off-policy exploration) and performs gradient descent steps on the loss, in our case the one in (3).

## 3 Method

The goal of CPDAG-GFN is to return multiple CPDAGs that approximate the true CPDAG underlying the data. The objective of using GFlowNet is to construct a posterior distribution over CPDAGs that will allow us to sample K high-reward CPDAGs. GFlowNet is suitable for this because of its capabilities as an amortized sampler, which allows for sampling during the training process and facilitates the exploration of high-reward CPDAGs throughout. Since the top K sampled high-scoring CPDAGs may often not align well with the true CPDAG underlying the data Koller & Friedman (2009), we incorporate a heuristic filter into our algorithm.

### 3.1 Heuristic Edge-Sparsity Filter

We introduce a heuristic filter that removes the $L$ least common edges among the top K sampled graphs at the end of training, while ensuring that each removal does not violate CPDAG properties in 1, thereby maintaining the graph's validity as a CPDAG. $L$ is a hyperparameter. This approach is motivated by our observation that edges consistently appearing across top K high-scoring models tend to reflect shared patterns in the true underlying graph. This observation led us to hypothesize that high-scoring graphs may share common edge features with the true CPDAG. The filter targets edges that do not consistently appear across models, which may mitigate the presence of spurious edges in the sampled graphs.

### 3.2 GFlowNet setup

The setup for GFlowNet includes defining a state space, a reward function, a graph neural network (GNN), and a loss function. For the loss function, we employ the trajectory balance function, which is covered in Section 2.2. We will now discuss each of these components in turn.

#### 3.2.1 State space

We define the state space to consist solely of CPDAGs, and the GFlowNet's action space as transitions from one CPDAG graph to the next.

Recall from Section 2.1 that a CPDAG may consist entirely of undirected edges, solely of directed edges, or a combination of both directed and undirected edges. We thus use the following three actions: add a directed edge, add an undirected edge, apply the `makeV` operator which transforms a graph structure from $x - y - z$ to $x \rightarrow y \leftarrow z$. In Appendix G, we show that these three actions are enough to construct any CPDAG in the search space.

In CPDAG-GFN, graphs are built starting from an edge-less graph $s_0$, with transitions to a new state achieved by applying one of the three actions mentioned above. For any given state, we limit allowable actions to those that lead to a new graph satisfying all the CPDAG properties outlined in Theorem 1 section 1. This ensures that the graph resulting from any permitted action will also be a CDPAG.

Additionally, we introduce a stop action, which serves as the termination point for a trajectory through the state space. If a stop action is sampled in $s_i$, we consider the state terminal and compute its reward $R(s_i)$ (see Figure 3).

#### 3.2.2 Reward function

Let $D$ represent a dataset of $N$ *i.i.d.* observations. Since we aim to sample CPDAGs that fit the data well, we define the reward as the score function $R(G) = score(G, D)$. We follow the definition of the score function by Koller & Friedman (2009):

$$score(G, D) = P(D \mid G)P(G), \tag{4}$$

where $P(G)$ is a structure prior which we set to be uniform. This configuration enables GFlowNet to explore the space of CPDAGs without any initial bias, and is a common choice (Eggeling et al., 2019; Koller & Friedman, 2009; Deleu et al., 2022).

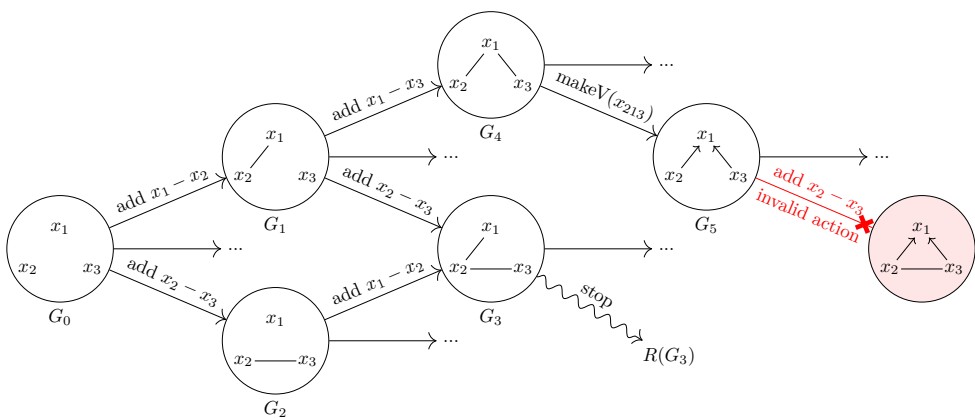

Figure 3: This figure illustrates the CPDAG construction process during GFlowNet training. It shows transitions from an initial edge-less graph state $G_0$ to subsequent states, distinguishing between valid actions that lead to new CPDAGs and invalid actions prohibited because they result in non-CPDAG states. Sampling a *stop* action at state $G_3$ concludes the trajectory, with the reward $R(G_3)$ then computed.

The marginal likelihood $P(D \mid G)$ can be calculated using any score-equivalent function such as the BGe score (Kuipers et al., 2014; Geiger & Heckerman, 1994), the BDe score (Heckerman et al., 1995; Chickering, 2013).[6] To obtain a score for a CPDAG, we first find a DAG belonging in the MEC that this CPDAG represents using an algorithm by Dor & Tarsi (1992), then assign that score to the CPDAG.

### 3.2.3 Parameterization with graph neural networks

In GFlowNet, we need to learn a policy that gives the probability of each state. Considering the exponential number of CPDAG states, it is impossible to store all these probabilities in a lookup table. Meanwhile, the training trajectories may not cover every state in the CPDAG space, which means we have to predict the probability for unseen states at test time. Following previous works Bengio et al. (2021); Deleu et al. (2022), we parameterize the policy with a neural network over the graph structure of the current state.

Considering the directed nature of CPDAGs, we employ a relational graph neural network (RGCN) Schlichtkrull et al. (2018) to encode the node representations, where directed and undirected edges are treated as different relations. For any node $u$, RGCN iteratively computes its representation $\boldsymbol{h}_u$ with the following message passing step

$$\boldsymbol{h}_u^{(t+1)} = \sigma \left( \sum_{r \in \mathcal{R}} \sum_{v \in \mathcal{N}_r(u)} \frac{1}{|\mathcal{N}_r(u)|} \boldsymbol{W}_r^{(t)} \boldsymbol{h}_v^{(t)} + \boldsymbol{W}_0^{(t)} \boldsymbol{h}_u^{(t)} \right) \tag{5}$$

where $\mathcal{R} = \{\text{directed}, \text{directed}^{-1}, \text{undirected}\}$ is the set of relations, $\mathcal{N}_r(u)$ is the set of nodes connected by relation $r$ from node $u$, $\boldsymbol{W}_r^{(t)}$ and $\boldsymbol{W}_0^{(t)}$ are learnable matrices and $\sigma$ is the activation function. The input embeddings $\boldsymbol{h}_u^{(0)}$ are initialized with one-hot embeddings to distinguish different nodes in the graph.

The three actions in Sec.3.2.1 correspond to link prediction and graph classification tasks on graph structure. Therefore, we decode the actions of adding edges with SimplE score function Kazemi & Poole (2018), a common choice for relational link prediction. Specifically, SimplE computes the following for a head node $u$, a relation $r$ and a tail node $v$

$$score(u, r, v) = (\boldsymbol{h}_u \odot \boldsymbol{r}_r)^\top \boldsymbol{K} \boldsymbol{h}_v \tag{6}$$

where $\boldsymbol{r}_r$ is a learnable embedding for relation $r$, $\odot$ is element-wise multiplication and $\boldsymbol{K}$ is an anti-diagonal identity matrix. Thanks to the asymmetry of the anti-diagonal kernel, SimplE can have different predictions

---

[6]Other score-equivalent functions – that is, those that assign the same score for any two Markov-equivalent graphs such as the Akaike information criterion (AIC) (Akaike, 1974), the Bayesian information criterion (BIC) (Schwarz, 1978), the minimum description length (MDL) (Rissanen, 1986), etc.

for $(u, r, v)$ and $(v, r, u)$, thereby is suitable for the action of adding directed edges in our model. To accommodate the action of adding a v-structure, we consider it as a link prediction problem between a set of two nodes $u_1$, $u_2$ and a collider node $v$, i.e. predicting $(\{u_1, u_2\}, r, v)$. The stop action is modeled by an MLP over the graph representation, which is obtained by a min pooling operation applied over all node representations in the graph.

## 4 Experiments

In this section, we evaluate the performance of our methods against state-of-the-art approaches by comparing the learned CPDAGs to the ground truth on both synthetic and real datasets.

### 4.1 Experimental evaluation

**Evaluation metric** We adopt the evaluation metrics used in prior work (Deleu et al., 2022; Lorch et al., 2021). We assess the performance of each algorithm using the Expected Structural Hamming Distance (E-SHD) — which measures the discrepancy between the inferred and the true CPDAGs (see appendix B for further detail), with lower E-SHD indicating better performance—and the Area Under the Receiver Operating Characteristic curve (AUROC), where a higher value signifies better performance. Additionally, we compute the average Structural Hamming Distance (SHD) between all pairs of CPDAGs in the sample, providing a measure of average dissimilarity across the CPDAGs. While this distance is not a formal evaluation metric, it gives us an indication of the diversity within the sampled graphs, with a higher average indicating greater dissimilarity.

**Baselines** To provide a comprehensive evaluation of our proposed method, we have selected baselines that use different approaches for approximating distributions over structural models. We include bootstrapping with PC (Spirtes et al., 2001) and Greedy Equivalence Search (GES) (Chickering, 2002b). Additionally, we employ DAG-GFN (Deleu et al., 2022), which leverages GFlowNet; DiBS (Lorch et al., 2021), which uses a variational inference approach; and MC3 (Madigan et al., 1995; Giudici & Castelo, 2003), representing MCMC-based methods.

### 4.2 Evaluation on synthetic data

In our experiments, we use the BGe score function as our reward function. Following Lorch et al. (2021), we generate ground truth networks using linear-Gaussian Bayesian networks, with their structures sampled according to the Erdős-Rényi (ER) model (ERDdS & R&wi, 1959) and the scale-free (SF) model (Barabási & Albert, 1999). These models were selected for their contrasting structural properties to ensure a diverse evaluation of CPDAG-GFN's adaptability and efficacy across different network topologies. In addition, we designed our experiments to include several scenarios: different observational data sizes ranging from small (100) to large (a million) to demonstrate scalability, varying levels of noise in the data from small (0.01) to moderate (0.1), and different network complexities with expected degrees of 1$d$, 2$d$, and 3$d$, where $d$ is the number of nodes, in which we set to d=10 in our experiments. For each experimental run, we sample K graphs, with K set to 100. Performance metrics (e.g. E-SHD and AUROC) are derived from 10 distinct datasets, each one generated from a unique Bayesian network.

**Discussion** Our experimental results (Figures 4 to 7) show that CPDAG-GFN performs competitively against the baselines across different scenarios. It might be worth clarifying that although CPDAG-GFN and DAG-GFN share a similar name and both use the GFlowNet framework, they are fundamentally different in their approach. DAG-GFN generates a distribution over DAGs and draws random samples, which are then converted to CPDAGs. In contrast, CPDAG-GFN leverages GFlowNet as an amortized sampler to directly sample top-K CPDAGs, which are further refined using a heuristic filter. Our experimental results show that CPDAG-GFN produces CPDAG candidates more aligned with the ground truth than DAG-GFN.

While B-GES demonstrates strong performance across various settings (Figures 5 to 7), the method relies on bootstrapping. The limitations of bootstrapping and scenarios in which they can fail do exist

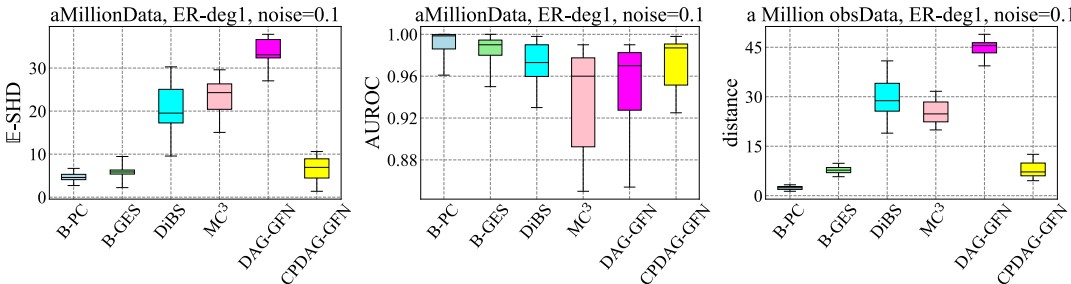

Figure 4: Comparison of E-SHD and AUROC metrics for a dataset generated from 1 million observations using a ground truth graph sampled from an Erdős-Rényi model (ER-deg1) with a noise level of 0.1. Lower E-SHD and higher AUROC are preferred. A higher distance in the third figure indicates greater dissimilarity among the graphs.

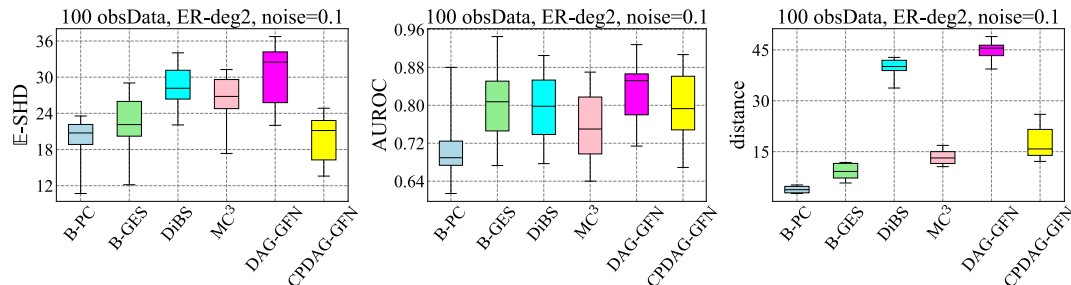

Figure 5: Comparison of E-SHD and AUROC metrics for a dataset generated from 100 observations using a ground truth graph sampled from an Erdős-Rényi model (ER-deg2) with a noise level of 0.1. Lower E-SHD and higher AUROC are preferred. A higher distance in the third figure indicates greater dissimilarity (e.g. more diversity) among the graphs.

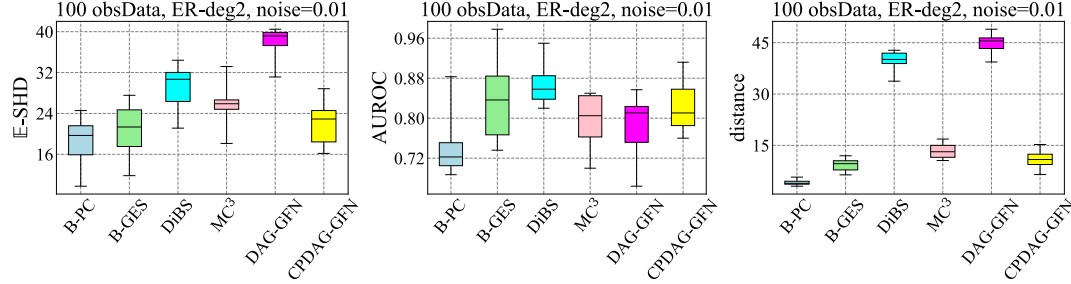

Figure 6: Comparison of E-SHD and AUROC metrics for a dataset generated from 100 observations using a ground truth graph sampled from an Erdős-Rényi model (ER-deg2) with a noise level of 0.01. Lower E-SHD and higher AUROC are preferred. A higher distance in the third figure indicates greater dissimilarity among the graphs.

Chernick (2011). Although these limitations are not specific to finding CPDAGs, they serve as a reminder of potential challenges with bootstrapping in certain contexts. While our experiments do not directly evaluate these scenarios, this perspective serves to suggest potential contexts where CPDAG-GFN may be preferred.

## 4.3 Real world data: Protein network from cell data

A well-known benchmark in structure learning is the causal protein-signaling network derived from data on 11 nodes representing proteins with 17 edges Sachs et al. (2005). In our experiments, we used the real-

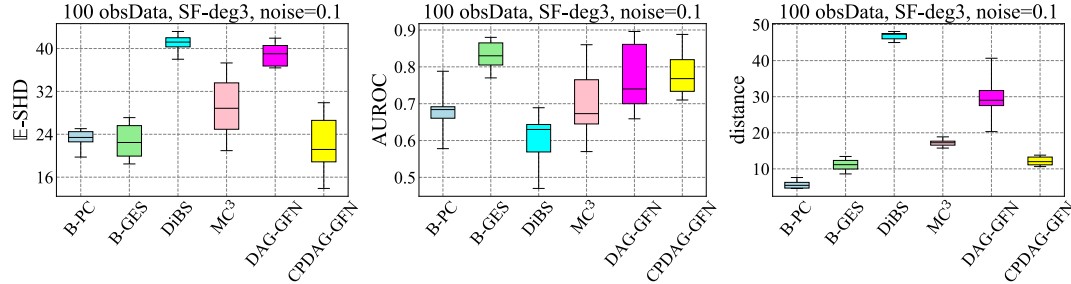

Figure 7: Comparison of E-SHD and AUROC metrics for a dataset generated from 100 observations using a ground truth graph sampled from a scale-free model (SF-deg3) with a noise level of 0.1. Lower E-SHD and higher AUROC are preferred. A higher distance in the third figure indicates greater diversity among the graphs.

| Method | $\mathbb{E}[\text{SHD}]$ | AUROC | $\mathbb{E}[\#\text{ Edges}]$ |
|---|---|---|---|
| $MC^3$ | $39.32 \pm 1.79$ | $0.645 \pm 0.041$ | $39.46 \pm 0.97$ |
| Bootstrap GES | $19.74 \pm 0.097$ | $0.751 \pm 0.011$ | $11.11 \pm 0.090$ |
| Bootstrap PC | $17.61 \pm 0.23$ | $0.728 \pm 0.019$ | $8.45 \pm 0.335$ |
| DiBS | $\mathbf{13.28 \pm 0.17}$ | $0.756 \pm 0.017$ | $11.47 \pm 0.34$ |
| DAG-GFN | $18.92 \pm 0.019$ | $0.658 \pm 0.019$ | $23.02 \pm 0.14$ |
| CPDAG-GFN | $16.61 \pm 0.65$ | $\mathbf{0.757 \pm 0.029}$ | $\mathbf{15.42 \pm 0.61}$ |

Table 1: Inference of protein signaling pathways from cytometry data (Sachs et al., 2005) Metrics are the mean ± SD of 10 experimental run

world protein network dataset provided in the supplementary materials of Sachs et al. (2005), consisting of $N = 854$ continuous observations. The results, presented in Table 1, compare the evaluation metrics E-SHD and AUROC of CPDAG-GFN against the baselines. For AUROC, a higher value is better, and for E-SHD, a lower value is better.

CPDAG-GFN demonstrates competitive performance against state-of-the-art methods, achieving an E-SHD of $16.61 \pm 0.65$. Notably, while DiBS achieves the lowest E-SHD, it predicts only 11.47 edges on average. In contrast, CPDAG-GFN's edge count of $15.42 \pm 0.81$ is closest to the ground truth network, which consists of 17 edges. Although AUROC values for DiBS, Bootstrap GES, and CPDAG-GFN are relatively close, CPDAG-GFN performs competitively by achieving a relatively low E-SHD, high AUROC, and an edge count closer to the ground truth compared to other baselines.

## 5 Related work

**Markov Chain Monte Carlo (MCMC)** Earlier papers that explored structure learning in the space of CPDAGs by means of MCMC include: Madigan et al. (1996) and Castelo & Perlman (2004). A more recent approach is by Castelletti et al. (2018). The paper focuses on learning sparse CPDAGs. To enhance the structure learning of these CPDAGs and ensure the graphs remain sparse, the authors introduce a sparsity constraint. This constraint limits the CPDAG space to a subspace where the number of edges does not exceed 1.5 times the number of nodes in the graph. While this approach can be effective, it can be a drawback in scenarios where the data-generating process is more complex and less sparse. In such cases, the constraint may bias the learning towards sparse graph that fail to capture more complex relationships within the data. Another draw back of using MCMC methods is their tendency to become trapped in a single mode of high probability (Syed, 2022), restricting their ability to explore diverse graph structures across different regions of high-probability modes. By using the GFlowNet approach instead of MCMC, the method we propose mitigates being confined to a single mode.

**Point estimate methods** Two widely known point estimate methods for computing a CPDAG from observational data are constraint-based methods and score-based methods. Constraint-based methods, such as the PC (Peter and Clark) algorithm and the Fast Causal Inference (FCI) algorithm (Spirtes et al., 2001), rely on conditional independence tests to identify the CPDAG that represents causal structures consistent with a given dataset (Eberhardt, 2017). In contrast, score-based methods, such as the Greedy Equivalence Search (GES; Chickering (2002b)), rely on a score function. These methods traverse the space of CPDAGs, assigning scores to graphs to measure their fit to the data. At each step, an edge is either added, removed, or reversed. As the name suggests, the search is greedy, choosing the state with the highest score to progress.

**Amortized sampling methods** Recent approaches to Bayesian structure learning that approximate the posterior distribution over DAGs include DAG-GFN (Deleu et al., 2022) and DiBS (Lorch et al., 2021), both operate within the space of DAGs. The DiBS method employs a variational approach, representing DAGs in a continuous latent space to facilitate the learning of the posterior distribution over network structures. Conversely, the DAG-GFN method uses GFlowNet to achieve a posterior distribution over DAGs where probabilities are approximately proportional to assigned rewards. CPDAGs can be obtained by converting the sampled DAGs to CPDAGs.

## 6 Conclusion

We have introduced a novel method for learning CPDAG candidates underlying observational data using GFlowNet. To better align the candidates with the ground truth, we applied a heuristic filter by removing the least common edges from the sample. Unlike traditional methods that produce a single CPDAG, our approach generates multiple CPDAG candidate structures by sampling directly from CPDAG distributions. In future work, we aim to explore the integration of domain-specific knowledge into the CPDAG-GFN algorithm. This could involve incorporating Bayesian priors to guide the sampling process toward more plausible causal structures, or developing heuristic-based constraints that reflect expert knowledge. By further incorporating specific causal hypotheses within the model, we hope to improve the accuracy and relevance of the generated CPDAGs, particularly in applications where expert knowledge is available.

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

## A   Sufficiency of actions to traverse CPDAG search spaces

In this section, we demonstrate that the three traversal actions—adding a directed edge, adding an undirected edge, and applying 'make V' operators—are sufficient to reach any CPDAG within the search space. To do so, we introduce the following two propositions:

**Proposition 2** (Andersson et al. (1997)). *Let $G$ and $H$ be two CPDAGs graphs with the same vertex set $V$. Then there exists a finite sequence $G \equiv G_1, \ldots, G_k \equiv H$ of CPDAGs with vertex set $V$ such that each consecutive pair of $G_i, G_{i+1}$ differs by one of the three traversal traversal actions: add directed edge, add undirected edge, or makeV.*

Building upon the aforementioned proposition, we present the following proposition:

**Proposition 3.** *For any essential $G$, there exists a sequence of graphs $G_0, \ldots, G_n$ such that:*

- *$G_0$ is the graph with no edges, and $G_n = G$.*
- *Each $G_i$ is a CPDAG.*
- *For each $i$, $G_{i+1}$ can be obtained from $G_i$ by one of the operators: add directed edge, add undirected edge, or makeV.*

*Proof.* This is immediate from the construction in the proof of Andersson's Proposition 4.5, which provides an algorithm for constructing such a sequence for any $G$. Note that the statement of Andersson's Proposition 4.5 alone does not imply this result, as it only guarantees a sequence $G_i$ such that for each $i$, either $G_i$ is obtained from $G_{i+1}$ by one of the operators, or $G_{i+1}$ is obtained from $G_i$ by one of the operators. $\square$

Proposition 3 above asserts that we can traverse the entire search space of CPDAGs starting from the empty graph $G_0$ and using only the three traversal actions (add a directed edge, add an undirected edge, or makeV operation), i.e., every state is reachable from the initial state. Each traversal action either increases the number of edges or does not change the number of edges while increasing the number of directed edges, which implies the search space is acyclic.

## B   Evaluation metric

We adopt the evaluation metrics used in prior work (Deleu et al., 2022; Lorch et al., 2021). We assess the performance of each algorithm using two primary measures: the Expected Structural Hamming Distance (E-SHD) to the ground truth graph $\mathcal{G}^*$, and the Area Under the Receiver Operating Characteristic Curve (AUROC). Intuitively, the AUROC measures the ability of the algorithms to accurately identify the presence or absence of edges compared to the ground truth. The higher the metric the better. We follow the methodology described in Deleu et al. (2022) to calculate the AUROC.

Following the methodology of (Deleu et al., 2022), the E-SHD to the ground truth graph ($\mathcal{G}^*$) is defined as:

$$\text{E-SHD} \approx \frac{1}{n} \sum_{k=1}^{n} \text{SHD}(\mathcal{G}_k, \mathcal{G}^*)$$

where $n$ represents the number of unique CPDAGs, and $\mathcal{G}_k$ denotes a CPDAG. The $\text{SHD}(\mathcal{G}, \mathcal{G}^*)$ counts the number of edge changes that separate the learned CPDAGs $\mathcal{G}_k$ from the ground truth $\mathcal{G}^*$ (Peters & Bühlmann, 2015; Lorch et al., 2021).

## C    Additional experiments

The ground-truth graphs are sampled according to an Erdos-Rényi model with average degrees equal to 1, 2, and 3, respectively, denoted ER-deg1, ER-deg2, and ER-deg3. Each experiments is ran between 7 to 9 seeds. The DAG-GFN was run using the publicly available code from Deleu et al. (2022).

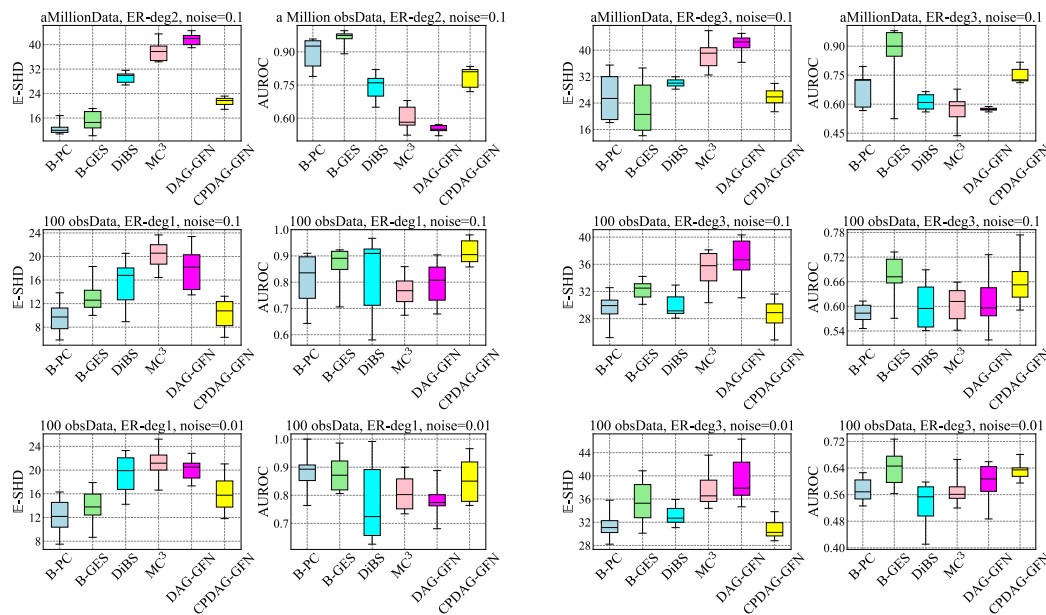

For the following experiments, the ground-truth graphs are sampled according to an scale-free model with average degrees equal to 1 and 2 respectively, denoted SF-deg1 and SF-deg2.

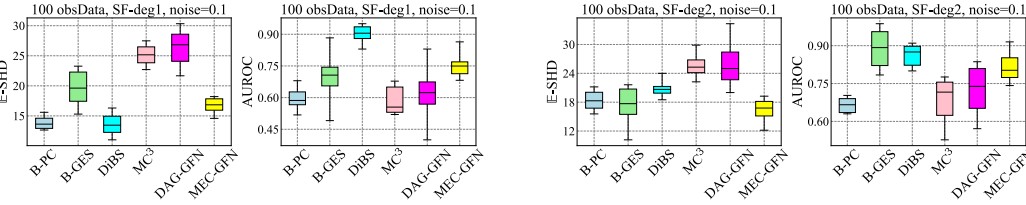

## D  DAG vs CPDAG comparison

To ensure a fair comparison, both DAG-GFN and CPDAG-GFN were implemented with the same setup, including the same optimizer, learning rate, neural network architecture, and dataset. We used the trajectory balance (TB) loss and BGe score function to train both models. Comparison of E-SHD and AUROC metrics for a dataset generated from large (e.g. 1 million) and small (e.g. 100) observations using a ground truth DAG sampled from an Erdős-Rényi model (ER-deg1) and scale-free model.

Since there are two important differences between Deleu et al. (2022)'s DAG-GFN and our work, namely the top-K sampling evaluation and the heuristic filtering, we compare four settings:

1. DAG-GFN with top-K sampling (denoted topK-DAG) during training versus CPDAG-GFN with top-K sampling (denoted topK-cpdag) during training.

2. DAG-GFN with $K$ samples after training (denoted rand-DAG) versus CPDAG-GFN with $K$ samples (denoted rand-cpdag) after training.

3. The same as in 1, but with the filter applied.

4. The same as in 2, but with the filter applied.

Baselines with a plus sign in front indicate the application of the least common edge removal technique. Figures 8 and 9 show experiments conducted for 5-variable graphs, and Figures 10 and 11 for 10-variable graphs.

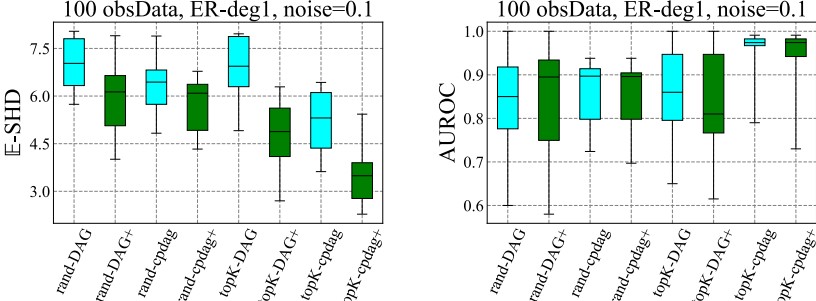

Figure 8: 5 variables: Lower E-SHD and higher AUROC indicate better performance compared to those with higher E-SHD and lower AUROC.

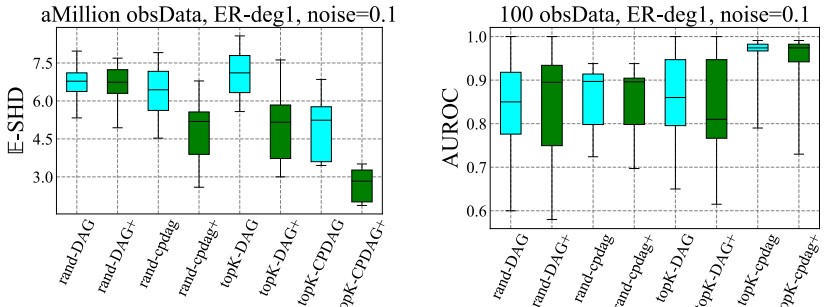

Figure 9: 5 variables: Lower E-SHD and higher AUROC indicate better performance compared to those with higher E-SHD and lower AUROC.

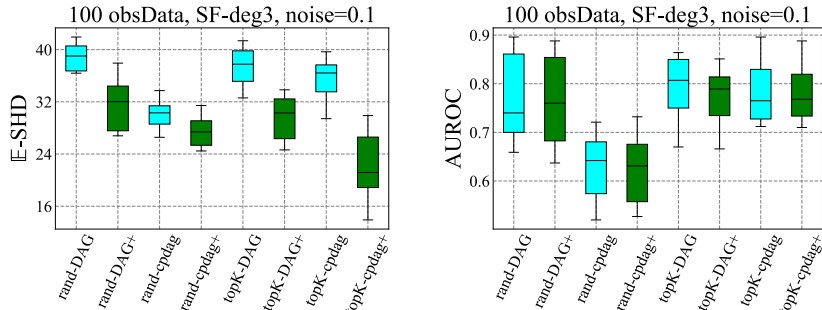

Figure 10: 10 variables: Lower E-SHD and higher AUROC indicate better performance compared to those with higher E-SHD and lower AUROC.

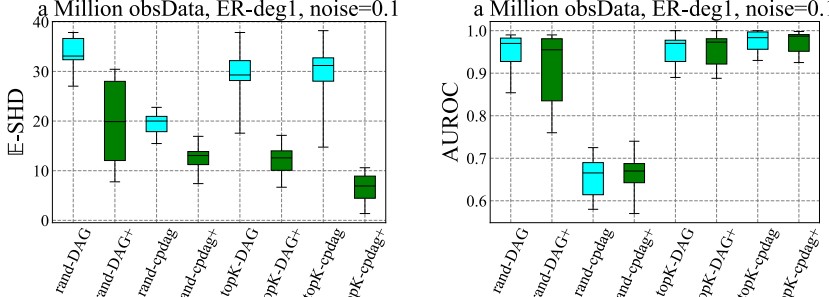

Figure 11: 10 variables: Lower E-SHD and higher AUROC indicate better performance compared to those with higher E-SHD and lower AUROC.

# E   Impact of Edge Removal on Baseline Methods

In this appendix, we investigate the impact of the heuristic filter (least common edge removal) on baseline methods using the same setup as in the main experiment (10 variables). In Figures 12, 13, 14, and 15, we compare baseline performance with (blue boxes) and without (green boxes) the removal technique.

While the edge removal technique leads to noticeable improvements for the baselines, as shown in the E-SHD figures, these gains are not always as substantial as those observed with our method. The bootstrap method showed the least improvement, suggesting that removing infrequent edges minimally impacts bootstrapped graphs, likely due to the initial similarity of the graphs, as indicated in the distance plots (Figures 5 to 7).

Our method consistently performs competitively across different settings compared to the baselines with the edge removal technique applied (blue box). This outcome shows that even with the addition of removing the least common edges, our method performs as well as or better than other baselines. In the figures below, baselines with a plus sign indicate the application of the least common edge removal technique. Moreover, topK-DAG represents K unique DAGs sampled from a posterior distribution over DAGs using GFlowNet, and then converting it to CPDAGs.

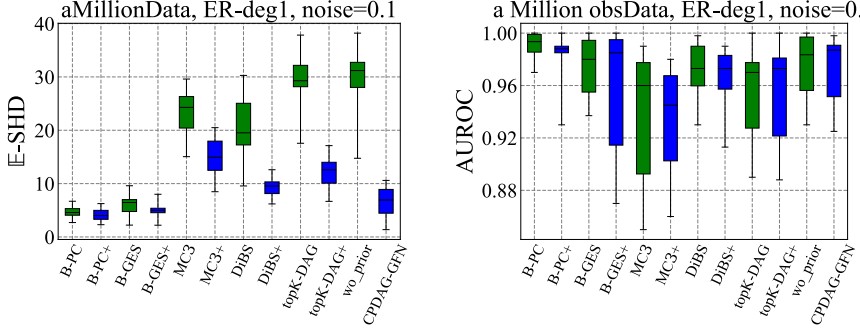

Figure 12: Baselines with lower E-SHD and higher AUROC indicate better performance compared to those with higher E-SHD and lower AUROC. Baselines with a plus sign indicate the application of the least common edge removal technique, and wo$_{prior}$ $represents CPDAG - GFN without the heuristic filter.$

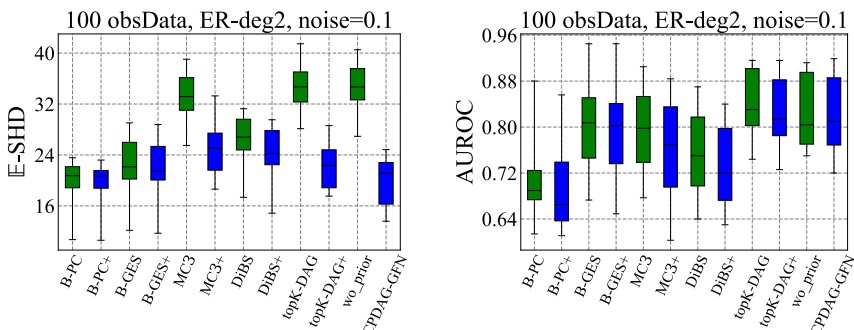

Figure 13: Lower E-SHD and higher AUROC indicate better performance compared to those with higher E-SHD and lower AUROC. Baselines with a plus sign indicate the application of the least common edge removal technique, and wo$_{prior}$ $represents CPDAG - GFN without the heuristic filter.$

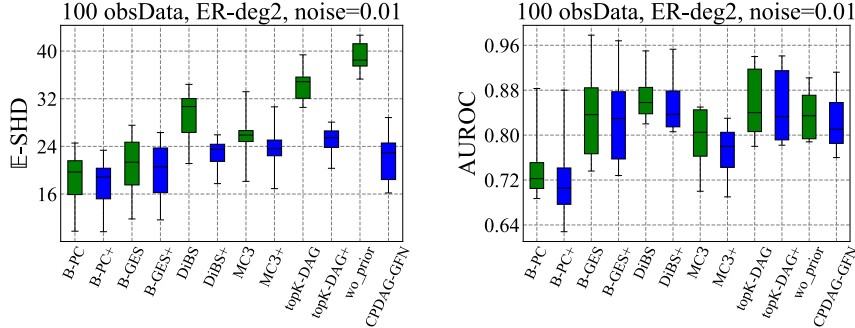

Figure 14: Lower E-SHD and higher AUROC indicate better performance compared to those with higher E-SHD and lower AUROC.Baselines with a plus sign indicate the application of the least common edge removal technique, and wo$_p$riorrepresentsCPDAG − GFNwithouttheheuristicfilter.

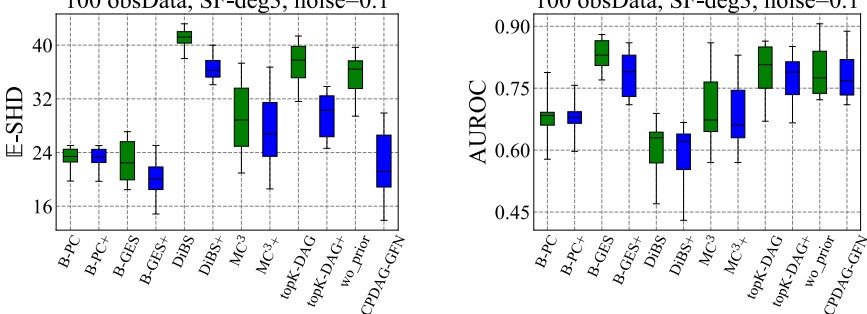

Figure 15: Lower E-SHD and higher AUROC indicate better performance compared to those with higher E-SHD and lower AUROC. Baselines with a plus sign indicate the application of the least common edge removal technique, and wo$_p$riorrepresentsCPDAG − GFNwithouttheheuristicfilter.

## F  Hyperparameter L analysis

In this section, we study how our method performs as we vary L.

**Experimental setup:** In the plots below, a unique Bayesian network was generated using different random seeds to create the data. This data was then used as input to our CPDAG-GFN algorithm to learn 100 CPDAG candidates. We computed E-SHD, AUROC, and distance metrics using these 100 CPDAGs for varying values of L. The results are presented in the plots below. All plots were generated using data from 1 million observations, ER-deg1, noise=0.1, and 10 variable setting.

**Observation:** The plots start at $L = 0$ (e.g., no least common edge is removed). As the hyperparameter $L$ increases, E-SHD initially decreases but begins to rise beyond a certain point (around $L = 40$). Similarly, AUROC starts off high but begins to drop as $L$ exceeds a specific threshold. On the other hand, the distance plot demonstrates an overall decreasing trend, with a noticeable bump in the purple scatter plot between $L = 30$ and $L = 40$. This suggests that removing certain edges may have increased the diversity of the graphs in the sample. Despite this bump, the general trend is downward, as expected, since removing the least common edges typically leads to greater similarity among the graphs, resulting in a decrease in the distance metric.

A suitable range for $L$ would be one that gives low E-SHD and high AUROC, and sufficient diversity in the generated graphs. Having said that, $L$ values ranging between 30 and 40 would be a reasonable choice.

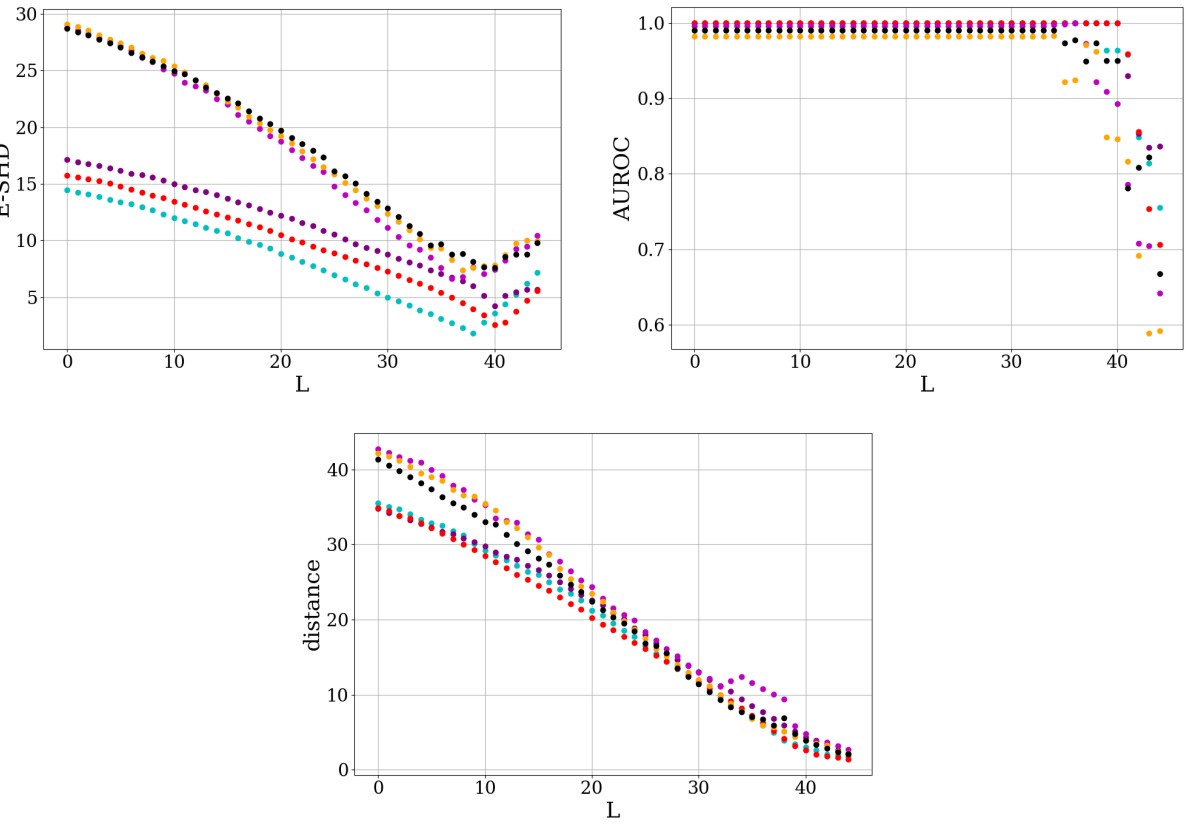

Figure 16: Lower E-SHD and higher AUROC indicate better performance.

# G    Defining the Mask Over Actions with CPDAG constraints

In this section, we discuss how we enforce theorem 1 of the CPDAG at each state.

To enforce the directed cycle constraint, we leverage the adjacency matrix, transitive closure, and outer product as inspired by Deleu et al. (2022) appendix C. This process generates a mask matrix, $\text{Mask}_{\text{matrix}}$, with entries of 0 or 1. We adopt the convention that entry (row, col) in the $\text{Mask}_{\text{matrix}}$ equal to 0 indicates that the directed edge row $\rightarrow$ col will not create a directed cycle, while an entry of 1 means it will.

We identify the indices of all zero entries (a,b) in $\text{Mask}_{\text{matrix}}$. Each of these indices is fed into the flow chart[7], which carries out the remaining constraints in theorem 1. The flowchart outputs 0 (indicating the action is allowed) or 1 (indicating the action is forbidden), and the $\text{Mask}_{\text{actions}}$ array is updated accordingly.

The purpose of $\text{Mask}_{\text{actions}}$ is to filter out all invalid actions so that only valid actions are sampled. Once a valid action is sampled and applied to generate a new CPDAG, the adjacency matrix, transitive closure, and mask matrix are updated to reflect the new state. Consequently, $\text{Mask}_{\text{actions}}$ is also updated accordingly.

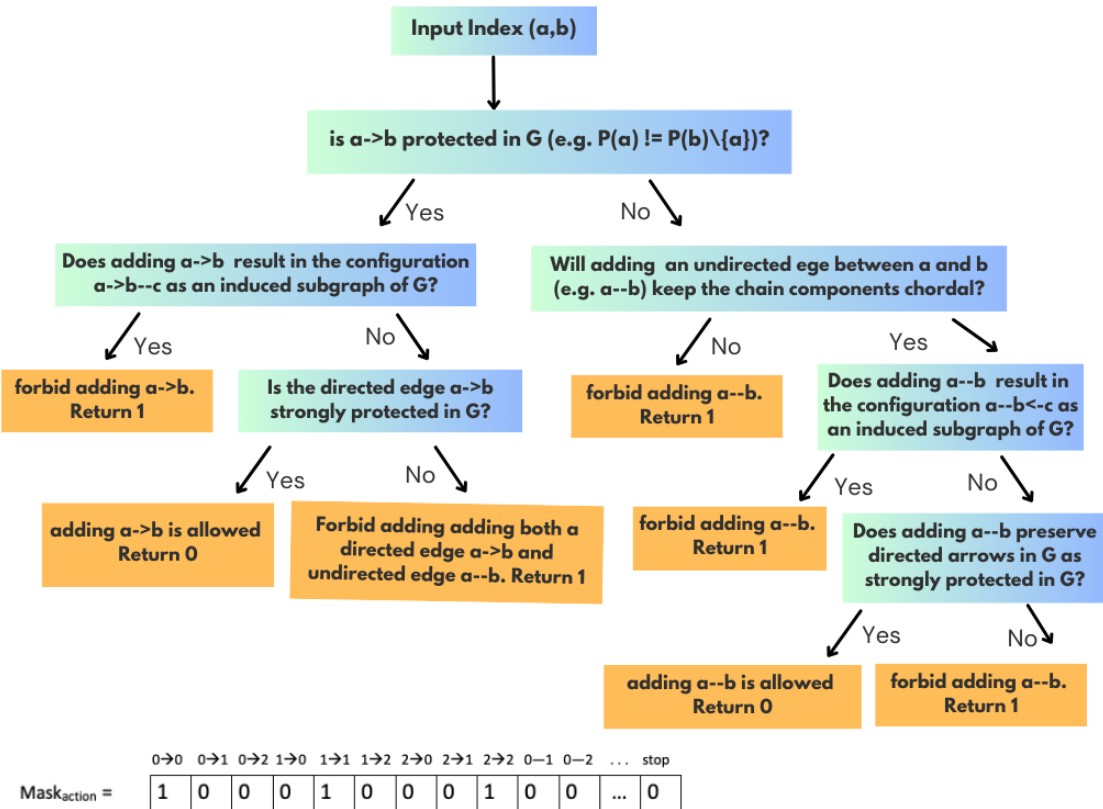

Figure 17: a) Flowchart illustrating how actions are validated and filtered at each state using the constraints outlined in Theorem 1. The input index(a,b) are the zero entries in the $\text{Mask}_{\text{matrix}}$. Here, $a \rightarrow b$ denotes a directed edge between node a and b in graph G, while $a - -b$ denotes an undirected edge between them. b) The $\text{Mask}_{\text{actions}}$ array entries are updated based on the flowchart's output, with 0 indicating a valid action and 1 indicating an invalid action.

---

[7]Please refer to Andersson et al. (1997) for further foundational and implementation details.

