# OpenReview forum: "Learning Equivalence Classes of Bayesian Network Structures with GFlowNet"
_TMLR — Rejected by TMLR_

### Review · Reviewer_2QW8 · 2024-09-23

**Summary Of Contributions:**

This paper proposes an approach for Bayesian network structure learning that returns Markov equivalence classes instead of Bayesian networks (which are DAGs). The paper proposes to do this with GFlowNets to optimize over the space of CPDAGs, which are a representation of Markov equivalence classes. The authors provide experimental evidence that using GFlowNets to optimize over CPDAGs does better than doing so over DAGs.

**Audience:**

Yes

**Claims And Evidence:**

No

**Requested Changes:**

Main requests:
Please address weaknesses 1, 2, and 3 above.

Other requests:
A)
> "We evaluate our method and baselines on a subset of this dataset, consisting of N = 852 continuous observations."

Please specify in the paper (appendix is fine) exactly how this subsetting was done, and/or better yet, organize the code repo to make this clear.

B) Why no mention of, nor comparison with, methods like NOTEARS and its descendants? I would likely reach for NOTEARS in each of the settings in the experiments section.

C) "CPDAG-GFN maintains a balance of a relatively low E-SHD, a comparatively high AUROC, and an edge count closer to the ground truth, indicating its overall competitive performance."

If you're going to claim this, it would be desirable to plot (E-SHD, AUROC)  scores across different train-test splits and across different methods to show that CPDAG-GFN is on the Pareto frontier.

D) For Table 1, please boldify the winner(s) of the AUROC column. Also, consider changing the definition of the E[# edges] column (eg mean squared error in number of estimated edges) so that either higher is better or lower is better. Or better yet, instead of AUROC and E[# edges], just show precision, recall, and F1 for edge recovery.

UPDATE 2024-12-14:

- The revision addresses Weakness 2 by removing the incorrect claim, although it leaves questions about the motivation for the proposed method. The revision addresses (A), but not (B), (C), and (D).

- After reading the revision and replies, I am even more confused about the underlying motivation for the proposed method. I would request two additional clarifications on (E) and (F).

(E) With respect to DAG-GFN, in their reply to the review, the authors state: "Thus when training DAG-GFlowNet to approximate completion, sparse DAGs within the same MEC(s) may be assigned relatively high scores, resulting in a high probability of sampling multiple DAGs from the same MEC(s). When these sampled DAGs are converted to CPDAG(s), they will map to the same CPDAG(s) which introduces redundancy in the sample."

It's not clear that this is a problem. That sparse CPDAGs correspond to MECs with a larger number of DAGs means that DAG-based methods will tend to assign greater posterior probability on sparse CPDAGs. This phenomenon has a regularizing effect, and it's not clear why this should be avoided, and instead preferred for the proposed sparsity-inducing heuristic, which seems less principled.

(F) With respect to B-GES, the Discussion section claims that, "While B-GES shows competitive performance across various settings (Figures 5 to 7), its dependence on resampling can introduce bias and potentially obscure subtle data patterns. In contrast, CPDAG-GFN uses the entire dataset in each run, ensuring that every data point contributes to model training and preserving the fidelity of the original data."

As far as I could tell, the paper offered no empirical examples or evidence that this is an actual issue for B-GES, and that CPDAG-GFN improves performance in such scenarios. Perhaps one of the synthetic or real data experiments corresponds to such a scenario? If so, explaining this in the relevant experiment section would be helpful.

**Strengths And Weaknesses:**

Strengths

1. It addresses an important problem, and showing that, for GFlowNets, it's better to search over MECs than individual Bayesian networks will likely be of interest to those who develop methods for this problem.

2. Overall, the authors offered a fair amount of empirical evidence and provided code, so I have confidence in the reproducibility of their findings of the empirical advantages of this method over DAG-GFlowNets.

3. The DAG vs CPDAG ablation experiments in Appendix D were very helpful.

Weaknesses

1. The first reason for this work is questionable. "it is likely that our prior over graphs matters." It's not clear that CPDAG-GFN is beneficial for this, because it means a user must specify a prior over CPDAGs instead of DAGs, but in many cases users might prefer to specify the latter. It's also not clear that a uniform prior over CPDAGs is better than one over DAGs. The former means that DAGs in larger MECs are given lower prior probability: why does this make intuitive sense?

2. The second reason for this work appears poorly motivated. "Furthermore, if a MEC is large ... [footnote] This gets worse as the variable space increase in size; the number of sparse DAGs encoding the same conditional independencies grows in an exponential-like manner (He et al., 2015)." This is misleading. The ratio CPDAGs / DAGs converges to roughly 0.267 (Gillispie & Perlman, 2002). The number of sparse DAGs grows exponential-like, but so does the number of corresponding CPDAGs. Furthermore, this phenomenon intuitively limits the benefits of the proposed approach because "potential redundancy in the samples obtained from the DAG space" does not grow as the number of variables grow. (Granted, the experiments show benefits for CPDAGs, but the claimed reason for this phenomenon is not convincing.)

3. While I appreciate that the authors provided code, I looked over it and was a little bit disappointed. While it's good enough for reproducibility purposes, it's not at all easy to use for practitioners. Installation requirements and instructions do not exist. And train.py doesn't provide an API function or class that a user can provide their own data and obtain a CPDAG; it appears to just be a script that's intermixed with the authors' own research code for experiments in the paper.

---

> ### Author Response · Authors · 2024-12-30
> **Reviewer 2QW8 ReReUpdate 2024-12-14**
>
> >Re Weakness 2: Fair enough! But my above concerns still leave unanswered questions about why one shouldn't simply use DAG-GFN instead.
>
> If the goal is to learn CPDAG candidates that approximate the ground truth from observational data alone, our experimental results show that CPDAG-GFN outperforms DAG-GFN in this task. Additionally, recall CPDAG-GFN generates a posterior distribution over CPDAGs, and  DAG-GFN produces a distribution over DAGs. Therefore, if one wants to obtain a posterior distribution that directly captures the epistemic uncertainty over CPDAGs or directly sample CPDAGs from a posterior distribution, CPDAG-GFN would be the appropriate choice.
>
> > Re metrics: To clarify, I am asking for TPR/FPR or Precision/Recall curves, to see methods' performances as a function of sparsity. Presumably, because AUROC is the area under the TPR/FPR (ie ROC) curve, it is possible to plot the curves on which the AUROC values were computed?
>
> the short answer is No.  To clarify, the objective of our work is not focused on sparsity, and the AUROC plots in our study should not be interpreted as a function of sparsity, nor are they intended to evaluate sparsity. As explained in our manuscript, AUROC in our experiments evaluates CPDAG-GFN's performance across experimental settings involving multiple factors, such as noise levels, data sizes, and network complexities. Consequently, calculating TPR/FPR as you described will not result in a plot as a function of sparsity.
>
>
> >Re NOTEARS: It's true that NOTEARS will have diversity of 0, which is of course a bad thing, all things else being equal. I don't think that this makes NOTEARS ineligible for comparison; rather, it will be helpful to see how much (if any) sacrifice must be made on other metrics in order to increase diversity
>
> Your statement does not align with the way we have defined our distance (or you call it diversity) metric in our manuscript. To reiterate, the distance metric in our work is designed to evaluate the diversity among multiple graphs in a sample and does not include comparisons of a graph with itself. For NOTEARS, the distance metric is not zero but rather *not applicable*, as NOTEARS produces only a single graph.
>
> We would also like to address your justification for including NOTEARS, specifically the statement: *“It's true that NOTEARS will have diversity of 0...it will be helpful to see how much (if any) sacrifice must be made on other metrics in order to increase diversity.”* This justification is fundamentally incorrect as it assumes that altering other metrics can increase the diversity of NOTEARS. The diversity metric you are referencing (calculated by taking the SHD of the single graph returned by NOTEARS with respect to itself) will always give a diversity of 0 (e.g. cannot be increased), regardless of the values of other metrics.
>
>
> >Re B-GES
>
> Thanks for sharing your feedback on how you interpreted the paragraphs to help us understand how our paragraph could be misunderstood outside of its  intended purpose. As we mentioned earlier, the paragraph was not intended to make any empirical claims but to remind the reader of the potential drawback of bootstrapping. We will modify the manuscript to make this more clear.

---

### Review · Reviewer_2yLL · 2024-10-30

**Summary Of Contributions:**

This paper studies Bayesian network structure learning utilizing Generative Flow Networks. Specifically, the work studies a simplified setting, where we learn DAGs only up to a Markov equivalence class (i.e., CPDAG) to narrow down the space of possible DAGs describing the data. The authors propose a GFlowNet-based framework that approximates the posterior over CPDAGs. Empirical results showcase the effectiveness of the proposed approach in learning CPDAGs from observational data.

**Audience:**

Yes

**Claims And Evidence:**

Yes

**Requested Changes:**

## Requested Changes

- It would be good for the authors to elaborate more on the motivations behind why we should model CPDAGs rather than methods that explicitly model DAGs.
- It would strengthen the paper to include hyperparameter studies changing L

**Strengths And Weaknesses:**

## Strengths

- Overall, I think the use of GFlowNets for causal discovery is certainly an important and interesting research direction
- The empirical results show that modeling CPDAGs using GFlowNets and adding priors into the learning produces generated graphs that are much closer to the ground truth than other baseline methods, including DAG-GFN.
- Using a GNN parameterization and formalizing the three steps (adding directed, adding undirected, and make v-structure) as graph prediction tasks is interesting.

## Weaknesses

- This paper heavily relies on DAG-GFN (Deleu et al). DAG-GFN mainly targets DAG learning using the GFlowNet framework, whereas this work tries to model only CPDAGs.
- The assumption in this paper of learning equivalence classes is strictly weaker than DAG-GFN. However, I do not see the motivation for why learning equivalence classes would be better when we can use DAG-GFN to learn close to the ground-truth structure.
- There are no experiments studying how the method performs as we vary L. Is L a significant hyperparameter to improve DAG learning? Or is the effect minimal? It would be good to justify why the heuristic filtering component is necessary.
- How does this method enforce the constraints of the CPDAG at each state? It would be good to clarify this.

---

### Review · Reviewer_ZM6C · 2024-11-27

**Summary Of Contributions:**

This paper makes use of GFlowNets to learn an amortized sampler on the
posterior distribution of causal DAGs consistent with the observations.
This approach operates on the set of edge constructions for the CPDAG
directly mimicking an action space reminiscent of edge prediction algorithms

**Audience:**

Yes

**Broader Impact Concerns:**

No ethical concerns

**Claims And Evidence:**

Yes

**Requested Changes:**

It would be helpful to really highlight where this method outshines the baselines. Since right now, it's hard to understand why someone should prefer this method over simpler baselines that give better accuracy and similar diversity of draws.

**Strengths And Weaknesses:**

Strengths

This work is well-written and easy to follow. The technical contribution is novel, and the problem it sets out to solve is interesting. The theoretical results all look accurate with no obvious errors.

Weaknesses

The contributed method doesn't stand out too much against the other baselines. The method in particular seems very similar to Deleu et al. And while the differences are well explained, it seems the heuristic introduced does not feel like a significant contribution. It's also not clear why a change that was made to offer greater diversity over DAG-GFNs seems to have lead to a worse result. Some exploration for why that might be would greatly strengthen this paper.

---

### Comment · Action_Editor_KfDe · 2024-11-27
**The discussion periods started**

Dear all,
the discussion period started, I invite authors and reviewers to engage the discussion.

All the best

---

### Decision · Action_Editor_KfDe · 2025-01-08

**Recommendation:** Reject

**Comment:**

The main motivations supporting the decision are related to the issues raised two out of three reviewers, namely ZM6C and 2QW8.
Indeed, as they pointed out by motivating their opinion, this paper needs more work to highlight the reasons why the proposed approach outperforms other works. Indeed, at the current state the authors mus significantly improve the clarity of what developed with specific refrence to motivating why their approach compares favourably to existing literature. I also share the view that to add clarity a more systematic evaluation and comparison of their method to state of the art is needed. I think that solving these issues, and provided a deeper understanding and explanation of the results will make this manuscript a very welcome addition to the literature.

**Audience:**

I'm fully convinced that the topic presented and analyzed in this manuscript is of relevance for the TMLR audience. Indeed, the issue of equivalence classes in Bayesian networks structure learning is a fundamental research problem.

**Claims And Evidence:**

According to 2QW8 the manuscript lacks in fulfilling claims and evidence.
I went trhough the motivations of the final recommendation from 2QW8 and found it well argumened and convincing.
In particular, I share the same view concerning the paper's claims, its fundamental motivations, and its empirical evidence, and on how these three fundamental aspects are set together.

I report hereafter the arguments from 2QW8 with regards to fundamental motivations (Weaknesses 1 and 2, and Requests E and F). Indeed, the paper does not offer consistent and convincing explanations for the proposed method, and in which scenarios it is expected to outperform previous methods. For example, (see Request F), the paper suddenly claims in the Discussion that B-GES is worse in certain scenarios due to its bootstrapping procedure, but offers no intuition or empirical evidence for this claim. Without conceptual clarity, there is always the worry that the proposed method either performed well due to favorable hyperparameter tuning, or due to selection of data for experiments. Another major issue for empirical evidence on claims is that the number of comparison seems to be limited, while the motivation for not experimenting with NOTEARS is not convincing at all. Furthermore, the difference between CPDAG-GFN inferred networks and those learned by other methods is not sufficiently presented and discussed. (See 2QW8  request for Precision/Recall (or TPR/FPR) and Pareto plots). I also agree on the third issue raised by 2QW8, that CPDAG-GFN doesn't have substantial across-the-board superiority is fine for acceptance, but only if the paper offers useful insight about when and why different methods do better in different scenarios. Such experiments and analysis are missing. Notably, there are no experiments or analyses that answer how different methods perform at different true and inferred sparsity levels, and only a single real data experiment. So it's hard to draw conclusions about when CPDAGs and when heuristic filtering are or are not helpful.

**Resubmission Of Major Revision:**

The authors may consider submitting a major revision at a later time.